# Multi-intention Inverse Q-learning for Interpretable Behavior Representation

**Hao Zhu**                                                    *zhuh@cs.uni-freiburg.de*
*Department of Computer Science & IMBIT//BrainLinks-BrainTools, University of Freiburg*

**Brice De La Crompe**                          *brice.de.la.crompe@biologie.uni-freiburg.de*
*Institute of Biology III & IMBIT//BrainLinks-BrainTools, University of Freiburg*

**Gabriel Kalweit**                                          *kalweitg@cs.uni-freiburg.de*
*Department of Computer Science, University of Freiburg*
*Collaborative Research Institute Intelligent Oncology (CRIION)*

**Artur Schneider**                             *artur.schneider@biologie.uni-freiburg.de*
*Institute of Biology III & IMBIT//BrainLinks-BrainTools, University of Freiburg*

**Maria Kalweit**                                           *kalweitm@cs.uni-freiburg.de*
*Department of Computer Science, University of Freiburg*
*Collaborative Research Institute Intelligent Oncology (CRIION)*

**Ilka Diester**                                  *ilka.diester@biologie.uni-freiburg.de*
*Institute of Biology III & IMBIT//BrainLinks-BrainTools, University of Freiburg*
*Bernstein Center Freiburg*

**Joschka Boedecker**                                        *jboedeck@cs.uni-freiburg.de*
*Department of Computer Science & IMBIT//BrainLinks-BrainTools, University of Freiburg*
*Collaborative Research Institute Intelligent Oncology (CRIION)*

**Reviewed on OpenReview:** *https://openreview.net/forum?id=hrKHkmLUFk*

## Abstract

In advancing the understanding of natural decision-making processes, inverse reinforcement learning (IRL) methods have proven instrumental in reconstructing animal's intentions underlying complex behaviors. Given the recent development of a continuous-time multi-intention IRL framework, there has been persistent inquiry into inferring *discrete* time-varying rewards with IRL. To address this challenge, we introduce the class of *hierarchical inverse Q-learning* (HIQL) algorithms. Through an unsupervised learning process, HIQL divides expert trajectories into multiple intention segments, and solves the IRL problem independently for each. Applying HIQL to simulated experiments and several real animal behavior datasets, our approach outperforms current benchmarks in behavior prediction and produces interpretable reward functions. Our results suggest that the intention transition dynamics underlying complex decision-making behavior is better modeled by a step function instead of a smoothly varying function. This advancement holds promise for neuroscience and cognitive science, contributing to a deeper understanding of decision-making and uncovering underlying brain mechanisms.

## 1 Introduction

Characterizing decision-making behavior stands as a fundamental objective within the field of behavioral neuroscience (Niv, 2009; Wilson & Collins, 2019). Prior research has formulated a variety of mathematical

behavioral models across diverse tasks (Ashwood et al., 2022b; Beron et al., 2022), including generalized linear models and models based on reinforcement learning. These *forward models* facilitate the understanding and comparison of decision-making strategies employed by both human and animal subjects. Additionally, they offer a low-dimensional behavioral representation suitable for regression analysis with neural activities (Hattori et al., 2019; Hamaguchi et al., 2022). Forward models require an empirically defined reward function that guides subjects optimizing their behavior during decision-making. However, defining a comprehensive and suitable reward function can pose challenges in complex behavioral tasks (Alyahyay et al., 2023; Rosenberg et al., 2021). Inverse reinforcement learning (IRL) (Ng et al., 2000; Arora & Doshi, 2021) is a popular approach to recover a reward function that induces the observed behavior, assuming that the demonstrator was maximizing its long-term return. Along with the significant successes of IRL in autonomous driving (Kalweit et al., 2020; Nasernejad et al., 2023), robotics (Kumar et al., 2023; Chen et al., 2023), and healthcare domains (Coronato et al., 2020; Chan & van der Schaar, 2021), it appears to be emerging as a valuable tool for constructing mathematical behavior models in neuroscience research, as exemplified by Yamaguchi et al. (2018); Kwon et al. (2020); Alyahyay et al. (2023).

Classic IRL methods seek to identify a single, fixed reward function for a specific scenario. In contrast, Ashwood et al. (2022a) suggested that animal's goals can evolve over time due to factors like fatigue, satiation, and curiosity. Under this assumption, they proposed the dynamic inverse reinforcement learning (DIRL) framework, which parametrizes the animal's reward function as a smoothly time-varying linear combination of a small number of spatial reward maps, which are referred to as 'goal maps'. By assuming the existence of multiple goal maps with time-varying weights, DIRL allows the instantaneous reward function to vary *continuously* in time. This innovative framework achieved state-of-the-art performance in animal behavior prediction. Nevertheless, demands have emerged regarding an IRL framework incorporating *discrete* time-varying reward functions, particularly following the proposal by Ashwood et al. (2022b) that natural behaviors can be represented through a Markov chain characterized by alternating between discrete intentions.

To address this requirement, we propose the novel class of *hierarchical inverse Q-learning* (HIQL) algorithms, which extend the fixed-reward inverse Q-learning (IQL) framework from Kalweit et al. (2020) to solve multi-intention IRL problems. Based on the assumption that the intention transition dynamics follows a Markov process, HIQL integrates an expectation-maximization (EM) approach to first divide expert trajectories into multiple intention segments in an unsupervised manner, and then solve the IRL problem independently for each segment. We then compare the performance of HIQL and the state-of-the-art model DIRL (Ashwood et al., 2022a) in a simulated gridworld environment and a dataset of trajectories from real mice navigating a 127-node-labyrinth (Rosenberg et al., 2021). The results show that our HIQL algorithm outperforms DIRL in behavior prediction in both benchmarks, and produces interpretable reward functions. Finally, we applied HIQL in a real mice decision-making dataset from a dynamic two-armed bandit task (De La Crompe et al., 2023), and mathematically characterized exploitation and exploration behavior of animals during value-based decision-making.

## 2    Related work

Various approaches have been introduced to address multi-intention IRL problems. Notably, several frameworks based on parametric (Babes et al., 2011; Likmeta et al., 2021), or non-parametric (Choi & Kim, 2012; Bighashdel et al., 2021) approaches allow for learning from multiple agents with distinct reward functions. Based on the assumption that the expert sticks to the same intention through one episode, these methods iterate between clustering expert demonstrations according to different intentions and solving the IRL problem for each cluster. However, unlike HIQL, these frameworks do not accommodate the scenario where intentions might alternate within the same episode.

In contrast, several approaches formulate MI-IRL problem as finding the maximum-likelihood partition of each trajectory, where each segment was generated from different locally consistent reward functions. To solve the problem, Dimitrakakis & Rothkopf (2012); Michini & How (2012) extended the Bayesian IRL frameworks (Ramachandran & Amir, 2007) to multi-intention scenario by introducing a Dirichlet process prior on different reward functions. The resulting Dirichlet process mixture model (DPM) can be solved via

probabilistic inference, *e.g.*, with Markov chain Monte Carlo (Neal, 2000). These algorithms do not require the number of intentions as a hyperparameter due to the nonparametric nature of DPMs. However, because of the nonparametric setting, DPMs inadequately model the temporal persistence of states and often create redundant states and rapidly switch among them, which eventually leads to poor interpretability of the results (Fox, 2009). This problem was addressed by Surana & Srivastava (2014), where they extended the DPM to a sticky hierarchical Dirichlet process hidden Markov model. Nevertheless, solving the IRL problem as Bayesian inference is still computationally intensive, which could be intractable even for moderately sized finite-state IRL problems. Under the same problem formulation, Nguyen et al. (2015) proposed a probabilistic graphical model, generalizing the algorithm from Babes et al. (2011). This approach avoids the computationally intensive Bayesian inference problem, but is restricted to the case of linearly-solvable environments. Our parametric HIQL algorithm also adapts the 'trajectory segmentation + IRL' problem formulation, but on the other hand, avoids the aforementioned limitations present in the other approaches. Particularly, we propose that in the parametric setup, manually specifying the number of intentions allows the researchers to decide the level of abstraction on intentions according to their expertise and scientific interest, *i.e.*, whether to focus on the fine-grained difference between intentions (*cf.* §5.3).

The state-of-the-art framework, DIRL (Ashwood et al., 2022a), can be considered as an extension of the maximum entropy IRL algorithm (Ziebart et al., 2008; 2010) to non-stationary rewards, where the reward function is assumed to be parameterized as a time-varying linear combination of a small number of non-linear spatial reward maps with Gaussian random walk prior over weights. Under this assumption, the borderline between intentions in DIRL is blur, *i.e.*, the intention transition is a continuous process, instead of a step-like procedure. However, such continuous time-varying reward assumed by DIRL is contradicted by the findings from Ashwood et al. (2022b), where it was proposed that humans and animals switch between multiple *discrete* strategies during decision-making. Although in theory, one can try to approximate such step-like intention transition dynamics with DIRL by assigning a larger variance $\sigma$ to the Gaussian to make the random walk less smooth, we show empirically (§5.1) that there are only minor influence from this hyperparameter $\sigma$ on the learnt reward, indicating that it would be challenging for DIRL to capture such step-like intention transition function during natural decision-making behavior. Instead, HIQL assumes that the intention transition dynamics follows a Markov process, which is intrinsically discrete and aligns better to the discrete switching intentions than DIRL (*cf.* §5.1 and §5.2).

Last but not the least, most of the aforementioned algorithms are *model-based*, relying on a known transition dynamics of the environment, whereas in many scenarios, the environment model is unknown. As an improvement, HIQL can also perform *model-free* learning, enabling the application in a wider range of environments.

## 3 Background

**Markov decision processes.** A Markov decision process (MDP) can be denoted by a tuple $(\mathcal{S}, \mathcal{A}, P, r, \gamma)$, where $\mathcal{S}$ and $\mathcal{A}$ denote the state- and action-space, respectively; The function $P \colon \mathcal{S} \times \mathcal{A} \times \mathcal{S} \to [0, 1]$ is the state transition function with $P(s, a, s') = \mathbf{P}(s' \mid s, a)$ and $\mathbf{1}^T P(s, a, \cdot) = 1$; The function $r \colon \mathcal{S} \times \mathcal{A} \to \mathbf{R}$ defines the reward function, and $\gamma \in [0, 1]$ denotes the discount factor. Additionally, the function $\pi \colon \mathcal{S} \times \mathcal{A} \to [0, 1]$ with $\pi(s, a) = \mathbf{P}(a \mid s)$ and $\mathbf{1}^T \pi(s, \cdot) = 1$ is used to represent the policy according to which actions are selected in the MDP.

**Inverse Q-learning.** Given the set of expert demonstrations $\mathcal{D}$ in an MDP, where each trajectory $\xi \in \mathcal{D}$ is denoted with a sequence of state-action pairs: $\xi = \{(s_0, a_0), \dots, (s_n, a_n)\}$, the IRL problem consists in determining a reward function $r$ that best explains the observed expert behavior in the form of demonstrated trajectories. Note that both the word 'demonstration' and 'trajectory' will be used subsequently. The former refers to one action execution and state transition of the expert, while the latter corresponds to a sequence of actions and state transitions for an entire episode. Unfortunately, IRL is generally ill-posed because infinitely many reward functions are consistent with the expert's observed behavior. To resolve this issue, various approaches have been proposed (see Arora & Doshi (2021) for a detailed review). Particularly,

Kalweit et al. (2020) formulated the IRL problem as a maximum likelihood estimation (MLE) problem:

$$
\begin{aligned}
\text{maximize} \quad & \mathbf{E}_{\xi \sim \mathcal{D}} \left[ \log \mathbf{P} \left( \xi \mid \pi_r \right) \right] \\
\text{subject to} \quad & \pi_r(s, a) = \exp \left( Q(s, a) - \log \sum \exp Q(s, \cdot) \right), \text{ for all } s \in \mathcal{S}, \, a \in \mathcal{A} \\
& Q(s, a) = r(s, a) + \gamma \sum_{s' \in \mathcal{S}} P(s, a, s') \max_{a' \in \mathcal{A}} Q(s', a'), \text{ for all } s \in \mathcal{S}, \, a \in \mathcal{A},
\end{aligned}
\tag{1}
$$

where $r$ is the optimization variable and $\mathcal{D}$ is the problem data. The objective of problem (1) is maximized when the expert policy $\pi^{\mathrm{E}}(s, a)$ is equivalent to $\pi_r(s, a)$. Hence,

$$
\pi^{\mathrm{E}}(s, a) = \frac{\exp(Q(s, a))}{\sum_{a' \in \mathcal{A}} \exp(Q(s, a'))} \implies Q(s, a) = Q(s, b) + \log(\pi^{\mathrm{E}}(s, a)) - \log(\pi^{\mathrm{E}}(s, b)),
\tag{2}
$$

for all actions $a \in \mathcal{A}$ and $b \in \mathcal{A}_{-a}$ where $\mathcal{A}_{-a} = \mathcal{A} \backslash \{a\}$. Using the Bellman optimality equation in (2), the immediate reward of action $a$ in state $s$ can be expressed by the immediate reward of some other action $b \in \mathcal{A}_{-a}$, the respective log-probabilities and future action-values (Kalweit et al., 2020):

$$
r(s, a) = \mu(s, a) + \frac{1}{\mathbf{card}(\mathcal{A}_{-a})} \sum_{b \in \mathcal{A}_{-a}} \left( r(s, b) - \mu(s, b) \right),
\tag{3}
$$

where $\mathbf{card}(\mathcal{A}_{-a})$ denotes the cardinality of $\mathcal{A}_{-a}$, and

$$
\mu(s, a) = \log(\pi^{\mathrm{E}}(s, a)) - \gamma \sum_{s' \in \mathcal{S}} P(s, a, s') \max_{a' \in \mathcal{A}} Q(s', a').
\tag{4}
$$

Let $\mathbf{card}(\mathcal{A}) = m$, for each state $s \in \mathcal{S}$, formulating (3) for all actions $a \in \mathcal{A}$ results in a system of linear equations about $r(s, \cdot)$:

$$
\begin{bmatrix}
1 & -\frac{1}{m-1} & \cdots & -\frac{1}{m-1} \\
-\frac{1}{m-1} & 1 & \cdots & -\frac{1}{m-1} \\
\vdots & \vdots & \ddots & \vdots \\
-\frac{1}{m-1} & -\frac{1}{m-1} & \cdots & 1
\end{bmatrix}
\begin{bmatrix}
r(s, a_1) \\
r(s, a_2) \\
\vdots \\
r(s, a_m)
\end{bmatrix}
=
\begin{bmatrix}
\mu(s, a_1) - \frac{1}{m-1} \sum_{b \in \mathcal{A}_{-a_1}} \mu(s, b) \\
\mu(s, a_2) - \frac{1}{m-1} \sum_{b \in \mathcal{A}_{-a_2}} \mu(s, b) \\
\vdots \\
\mu(s, a_m) - \frac{1}{m-1} \sum_{b \in \mathcal{A}_{-a_m}} \mu(s, b)
\end{bmatrix}.
\tag{5}
$$

Thus the unknown reward function $r$ can be obtained by solving (5) for all $s \in \mathcal{S}$ via least squares (Kalweit et al., 2020). This approach is called the *inverse action-value iteration* (IAVI) (Algorithm 1), a model-based algorithm that solves the IRL problem analytically in closed-form. When the transition dynamics $P$ is unknown, stochastic approximation can be applied to (3) and (4) such that $r$ can be obtained in a sampling-based manner (Kalweit et al., 2020). The resulting *inverse Q-learning* (IQL) algorithm (Algorithm 2) is a model-free extension of IAVI. The class of IQL algorithms solve the MDP underlying the demonstrated behavior only once, leading to a speedup of up to several orders of magnitude compared to the popular maximum entropy IRL algorithm (Ziebart et al., 2008) and some of its variants. In addition, it can accommodate arbitrary non-linear reward functions.

## 4 Hierarchical inverse Q-learning

We formulate the multi-intention IRL problem under the following assumptions:

**Assumption 4.1.** Each expert demonstration is generated according to the Boltzmann optimal policy under one of the reward functions in a $K$-dimensional finite set $\mathcal{R} = \{r_1, \ldots, r_K\}$, with each corresponding to one specific intention.

**Assumption 4.2.** The probability that one demonstration is generated under reward function $r \in \mathcal{R}$ is controlled by a Markov chain with initial state distribution $\Pi$ and transition matrix $\Lambda$, where the vector $\Pi$ and each row of the transition matrix $\Lambda_{i:}, i = 1, \ldots, K$ represent some probability distribution over the set $\mathcal{R}$, *i.e.*, $\Pi \in \Delta^{K-1}$ and $\Lambda_{i:} \in \Delta^{K-1}$, where $\Delta^{K-1} = \left\{ x \in \mathbf{R}^K \mid x \succeq 0, \mathbf{1}^T x = 1 \right\}$ is a $(K-1)$-dimensional probability simplex.

The resulting hierarchical decision process of an expert following these assumptions from the perspective of a researcher is depicted graphically in Figure 1, where the first two rows represent the MDP, and the last row represents the Markov chain for intention transition dynamics. Solving IRL problems on such decision network with parameter space $\Theta = \{\Pi, \Lambda, \mathcal{R}\}$ consists in determining 1) a set of reward functions, and 2) the reward function index for each demonstration that best jointly explain the observed expert behavior. An expectation-maximization (EM) algorithm can be devised to iteratively learn $\Theta$. For convenience, we introduce $\eta = \{z_0, \ldots, z_n\}$ to be the pre-

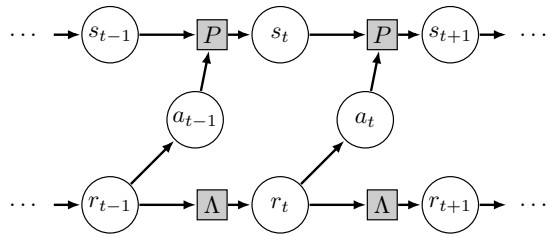

Figure 1: Graphical representation of expert's decision process.

dicted sequence of reward function indexes for trajectory $\xi \in \mathcal{D}$. Then each iteration of the EM process can be written as an MLE problem where the likelihood is obtained by marginalizing out the sequence of predicted reward function indexes $\eta$:

$$\text{maximize} \quad J\left(\Theta^+ \mid \Theta\right) = \mathbf{E}_{\xi \sim \mathcal{D}, \eta}\left[\log \mathbf{P}\left(\xi, \eta \mid \Theta^+\right)\right], \tag{6}$$

where $\Theta^+$ is the optimization variable and $\mathcal{D}, \Theta$ are the problem data.

**Theorem 4.3.** *Solving problem (6) is equivalent to solving a sequence of optimization problems:*

$$
\begin{aligned}
&\text{maximize (over } \Pi^+) && \mathbf{E}_{\xi \sim \mathcal{D}}\left[\sum_{i=1}^K \mathbf{P}(z_0 = i \mid \xi, \Theta) \log \Pi_i^+\right] \\
&\text{subject to} && \Pi^+ \succeq 0, \ \mathbf{1}^T \Pi^+ = 1,
\end{aligned} \tag{7}
$$

$$
\begin{aligned}
&\text{maximize (over } \Lambda^+) && \mathbf{E}_{\xi \sim \mathcal{D}}\left[\sum_{i=1}^K \sum_{j=1}^K \sum_{t=1}^n \mathbf{P}(z_{t-1} = i, z_t = j \mid \xi, \Theta) \log \Lambda_{ij}^+\right] \\
&\text{subject to} && \Lambda_{i:}^+ \succeq 0, \ \mathbf{1}^T \Lambda_{i:}^+ = 1, \quad i = 1, \ldots, K,
\end{aligned} \tag{8}
$$

*and*

$$
\begin{aligned}
&\text{maximize (over } r_i^+) && \mathbf{E}_{\xi \sim \mathcal{D}}\left[\sum_{t=0}^n \mathbf{P}(z_t = i \mid \xi, \Theta) \log \pi_{r_i^+}(s_t, a_t)\right] \\
&\text{subject to} && \pi_{r_i^+}(s, a) = \exp\left(Q(s, a) - \log \sum \exp Q(s, \cdot)\right), \text{ for all } s \in \mathcal{S}, \ a \in \mathcal{A} \\
& && Q(s, a) = r_i^+(s, a) + \gamma \sum_{s' \in \mathcal{S}} P(s, a, s') \max_{a' \in \mathcal{A}} Q(s', a'), \text{ for all } s \in \mathcal{S}, \ a \in \mathcal{A}.
\end{aligned} \tag{9}
$$

*Proof.* See Appendix A.1. □

In practice, to evaluate the objective functions of (7), (8) and (9), the Baum-Welch algorithm (Baum & Petrie, 1966) can be applied to obtain the required posterior probabilities $\mathbf{P}(z_t = i \mid \xi, \Theta)$ and $\mathbf{P}(z_{t-1} = i, z_t = j \mid \xi, \Theta)$ (*cf.* Appendix A.2). Then according to the Gibbs' inequality, the optimum of (7) and (8) are achieved by

$$\Pi_i^+ = \mathbf{E}_{\xi \sim \mathcal{D}}\left[\mathbf{P}(z_0 = i \mid \xi, \Theta)\right], \quad i = 1, \ldots, K, \tag{10}$$

and

$$\Lambda_{ij}^+ = \frac{\mathbf{E}_{\xi \sim \mathcal{D}, t}\left[\mathbf{P}(z_{t-1} = i, z_t = j \mid \xi, \Theta)\right]}{\mathbf{E}_{\xi \sim \mathcal{D}, t}\left[\mathbf{P}(z_{t-1} = i \mid \xi, \Theta)\right]}, \quad i = 1, \ldots, K, \quad j = 1, \ldots, K. \tag{11}$$

To solve problem (9), note that it has the same structure as (1), thus it can be addressed by the class of IQL algorithms (Algorithm 1 and 2) with slight adaptations. Specifically, for each $i = 1, \ldots, K$, a subset of expert demonstrations from $\mathcal{D}$ will be sampled w.r.t. the posterior probability $\mathbf{P}(z_t = i \mid \xi, \Theta)$, then either Algorithm 1 or Algorithm 2 (depending on whether the transition dynamics $P$ is available) will be applied to this subset of demonstrations so that $r_i^+$ can be obtained. Assembled, we call the above process of solving multi-intention IRL problems the *hierarchical inverse Q-learning* (HIQL) algorithm. The corresponding pseudo-code is listed in Algorithm 3.

# 5 Experiments and discussion

We evaluate the performance of HIQL on a simulated gridworld environment and a real mice behavior dataset obtained from a 127-node labyrinth navigation task (Rosenberg et al., 2021), and compare to DIRL (Ashwood et al., 2022a). We then show the potential of performing model-free learning and detecting exploration behavior with HIQL on a mice reversal-learning task (De La Crompe et al., 2023).

## 5.1 Gridworld benchmark

The simulated gridworld environment is a $5 \times 5$ map (Figure 2), where an agent can choose between going *up*, *down*, *left*, *right*, or to *stay* in place per time step. The expert starts at origin $(0, 0)$ and any of its actions can achieve the intended state with 90% probability, while the other 10% lead to random directions. It has two possible policies, where $\pi^{\text{goal}}$ prefers going to the destination $(4, 4)$ and $\pi^{\text{abandon}}$ prefers returning to origin $(0, 0)$. The expert starts an episode always with the $\pi^{\text{goal}}$ policy. While moving to the destination, the expert will encounter barriers '#' at some states. Every time the expert runs into a barrier, it has 30% probability of switching to a different policy. Each episode ends when the expert reaches either the origin or the destination, and at the 8th time step, if the episode has not finished, the expert will have a 50% probability of switching to $\pi^{\text{abandon}}$ (*cf.* Appendix C.1). Such intention transition dynamics of the simulated expert as described above is better aligned to a step-function, instead of a smoothly varying function.

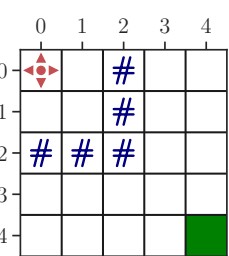

Figure 2: The gridworld environment.

We compared between the performance of HIAVI (the model-based variant of HIQL) and DIRL with 1 or 2 intention(s). Note that for the single intention case, HIAVI falls back to normal IAVI and DIRL falls back to maximum causal entropy IRL (Ziebart et al., 2010). Trying to enable DIRL to better capture such step-like intention transition dynamics, we varied the hyperparameter that controls the smoothness of the time-varying reward in DIRL — the variance $\sigma$ for the Gaussian of the random walk prior, from 0.01 to 10. A larger $\sigma$ value corresponds to a less smooth intention transition dynamics in DIRL. The whole expert demonstration dataset consisted of 1024 trajectories. All evaluated algorithms were fit to multiple sub-datasets with different number of expert trajectories, and each dataset was analyzed using a 5-fold cross-validation. We list the detailed information about model training in Appendix C.1.

In general, HIAVI outperformed DIRL in predicting the expert behavior as indicated by the log-likelihood on the test dataset (Figure 3a). Notice that HIAVI already had a better performance compared to DIRL even in the collapsed single intention model, indicating that IAVI serves as a better inner-loop (IRL problem) solver than maximum entropy IRL. This is probably because IAVI learns the unknown expert reward in closed-form by solving a system of linear equations instead of performing parameter estimation as maximum entropy IRL does, which introduces approximate error (Kalweit et al., 2020). Comparing the single- and multi-intention variants, both HIAVI and DIRL had an increase in prediction performance. Particularly, the HIAVI almost achieved the expert level as the number of trajectories increased, while the performance improvement for DIRL was only minor. Besides, different $\sigma$ values of DIRL seem not to have a significant influence on the DIRL performance when there is enough expert trajectories in the dataset.

Next we compared HIAVI and DIRL abilities in capturing the expert's intention transition dynamics (Figure 3b) and recovering the expert policy (Figure 3c). The analysis was performed based on the learnt policy using the whole dataset with all 1024 trajectories (with cross-validation applied). As a measure of performance in policy recovery, we used the expected value difference (EVD) metric (Levine et al., 2011). EVD is defined as the mean square error between the state-value under the true reward function for the expert policy and the state-value under the true reward for the optimal Boltzmann policy w.r.t. the learnt reward. It provides an estimation of the sub-optimality of the learnt policy under the true reward function. For HIAVI and DIRL with 2 intentions, the inferred intentions were assigned to the best-fit ground truth intentions, and the two learnt policy was evaluated under the corresponding true reward individually to obtain the respective state-values. Under the single-intention setup, since only one learnt policy could be

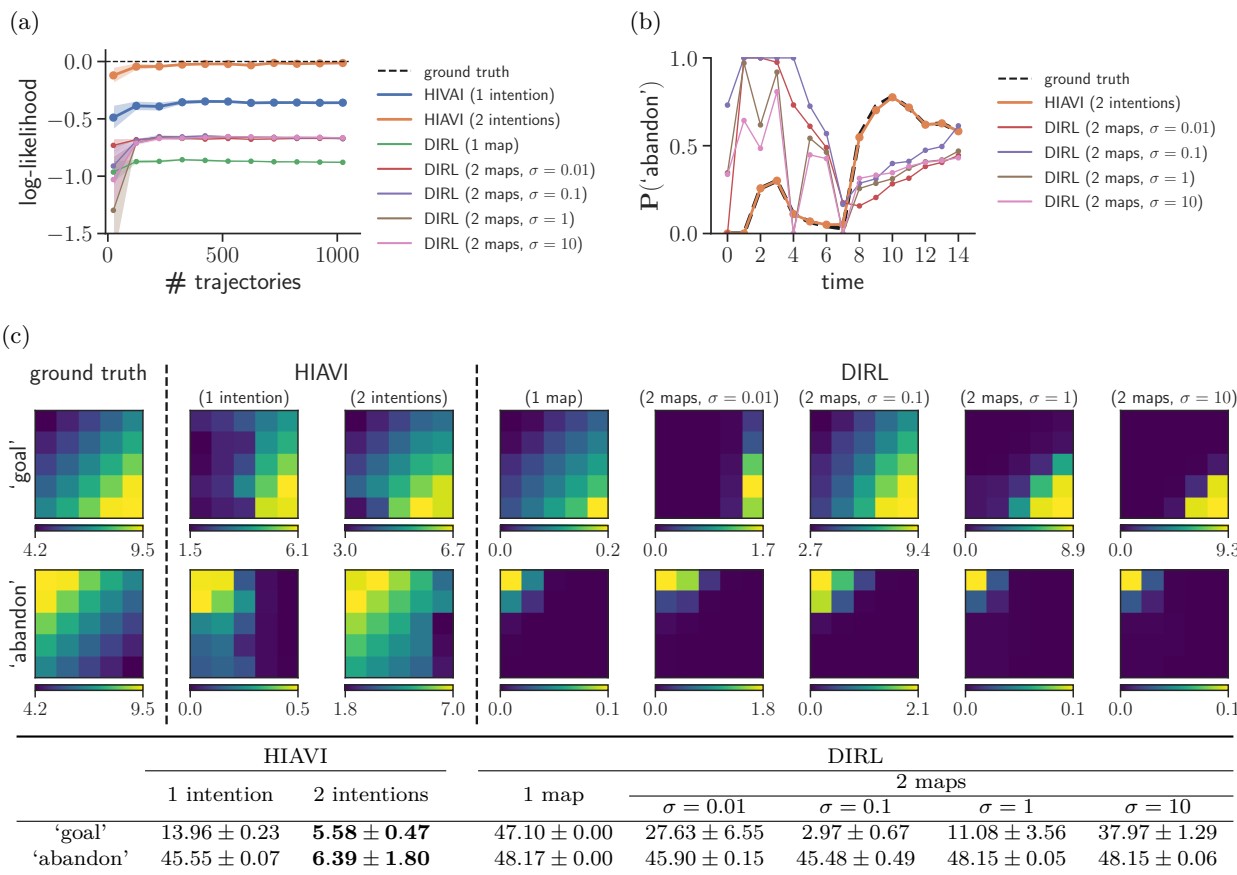

Figure 3: Results for the gridworld benchmark. (a) Comparison of HIAVI and DIRL on datasets with different number of expert trajectories, represented as log-likelihood on the test dataset (mean ± standard error, 5-fold cross-validation). (b) Predicted intention dynamics from HIAVI and DIRL using the outputs from the best cross-validation fold, represented as the posterior probability of the 'abandon' intention and averaged across all trajectories (mean ± standard error across 1024 trajectories). (c) Visualization of the ground truth and learnt state-value functions from the best cross-validation fold (top), and the corresponding EVDs (bottom, mean ± standard error, 5-fold cross-validation) from HIAVI and DIRL.

provided by the algorithms, the same output learnt policy was evaluated twice repeatedly under the different true reward functions, respectively. Aligning with the results indicated by Figure 3a, HIAVI had smaller EVD values than DIRL even under single-intention setup (Figure 3c). Particularly, the learnt policy from single-intention HIAVI exhibits a mixed characteristic of both 'goal' and 'thirsty' intentions but is more close to the former, whereas the learnt policy from single-intention DIRL deviates from both of the two intentions. This explains why HIAVI already had a better prediction log-likelihood than DIRL under the single-intention setup. Then by introducing the second intention, the best performance in recovering expert policy under both intentions was achieved by HIAVI ($5.58 \pm 0.47$ for $\pi^{\text{goal}}$ and $6.39 \pm 1.80$ for $\pi^{\text{abandon}}$). While by selecting an appropriate $\sigma$ value, DIRL could reconstruct the 'goal' policy at very high level (with a $2.97 \pm 0.67$ EVD when $\sigma = 0.1$), it struggled to learn the policy under 'abandon' intention. As a result, the HIAVI predicted posterior probability of the expert remaining in the 'abandon' intention at different time steps in a single episode matched exactly with the ground truth (Figure 3b), whereas DIRL seemed to overestimate this probability at the beginning (where the expert reached the barrier for the first time), and underestimate it after the 8th time step.

To sum up, the comparison between HIAVI and DIRL on the gridworld benchmark suggests that HIAVI can better capture a step-like intention transition dynamics and reconstruct the underlying expert policies

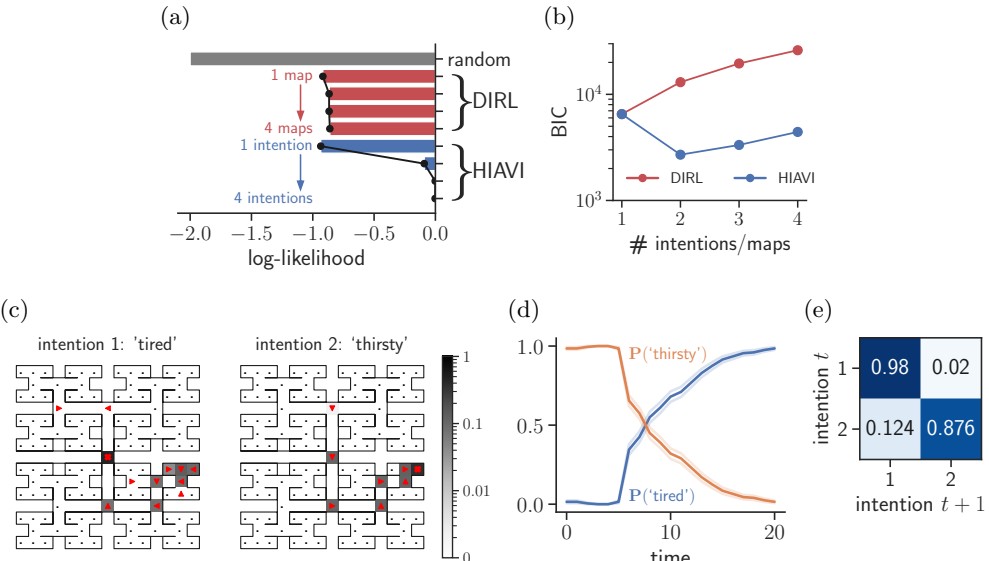

Figure 5: Results for the navigation benchmark of the water-restricted cohort. (a) Comparison of HIAVI, DIRL, and a random policy, represented as log-likelihood on the test dataset. (b) BIC as a function of the number of intentions in HIAVI. (c) Learnt policy (red arrows and crosses) in the environment and corresponding state occupancy (grey colormap) under different intentions. (d) Predicted intention dynamics from HIAVI, averaged across all 200 trajectories. Solid and shaded curves denote the mean and standard error across 200 trajectories. (e) Inferred intention transition matrix from HIAVI.

than DIRL. Although in theory one can try to approximate such step-function with DIRL by assigning a larger variance $\sigma$ to the Gaussian to make the random walk less smooth, it has been shown empirically in this experiment that it's not trivial in practice since the influence of this hyperparameter $\sigma$ on DIRL would only be limited.

## 5.2 Real-world mice navigation benchmark

The expert demonstrations in this real-world dataset were originally collected by Rosenberg et al. (2021) from a 127-node labyrinth navigation task. As shown in Figure 4, each black dot represents one state in the environment and the subjects can select from 4 actions: *left*, *right*, *reverse*, and *stay* at each state. One of the 127 states is occupied with water resource (the blue square), and the optimal path from the labyrinth entrance to the water port is depicted as blue line. Two cohorts of 10 mice moved freely in dark through the labyrinth over the course of 7 hours, where one cohort of animals was under water restriction while the other was not. Difference in water restriction condition resulted in different animal behavior between these two cohorts in the environment. For model evaluation, HIAVI and DIRL with varied intentions from 1 to 4 were trained independently on the water-restricted and water-unrestricted expert demonstration datasets (with 200 and 207 animal trajectories, respectively). 20% of the trajectories from each dataset were held out as a test set. We list the detailed information about trajectory preprocessing and model training in Appendix C.2.

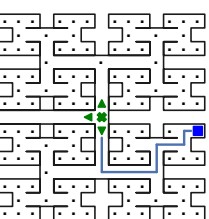

Figure 4: The labyrinth environment.

We first compared between HIAVI and DIRL on the trajectories collected from the group of water restricted animals. First of all, as the number of intentions increased, the log-likelihood value for HIAVI predictions increased significantly, whereas the improvement of DIRL performance was rather minor (Figure 5a). This

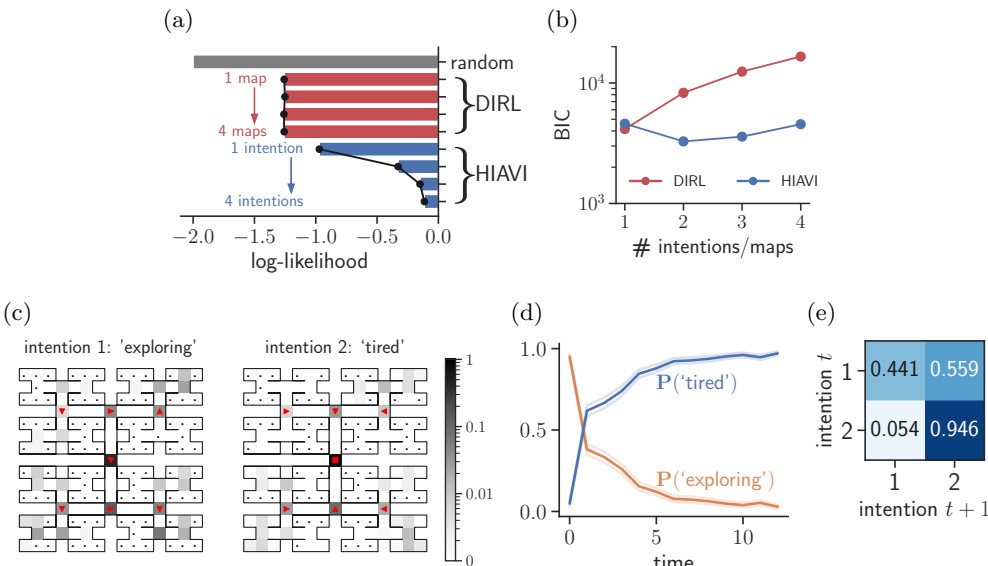

Figure 6: Results for the navigation benchmark of the water-unrestricted cohort. (a) Comparison of HIAVI, DIRL, and a random policy, represented as log-likelihood on the test dataset. (b) BIC as a function of the number of intentions in HIAVI. (c) Learnt policy (red arrows and crosses) in the environment and corresponding state occupancy (grey colormap) under different intentions. (d) Predicted intention dynamics from HIAVI, averaged across all 207 trajectories. Solid and shaded curves denote the mean and standard error across 207 trajectories. (e) Inferred intention transition matrix from HIAVI.

led to a distinguishable outperformance of HIAVI compared to DIRL in this benchmark. To check whether the better performance of HIAVI compared to DIRL was due to a larger number of papameters, we then calculated the Bayesian information criterion (BIC) for these two approaches (Figure 5b). It shows that the BIC values for HIAVI are consistently lower than those for DIRL when the number of intentions exceeds one. To analyze the recovered expert policies by HIAVI, we selected the number of intentions $K = 2$ where the lowest BIC value was achieved. The learnt mice policy under intention 1 ('tired') displays a preference of moving out from the water port towards the maze entrance and stay, while the policy under intention 2 ('thirsty') guides the mice directly to the water port along the optimal track. Correspondingly, under the 'tired' intention, the highest state occupancy was observed at the entrance of the environment, while under 'thirsty', it was observed at the water port (Figure 5c). To visualize the predicted intention transition dynamics, we computed the posterior probability over mice's intentions across all trajectories. The recovered temporal intention dynamics shows a high probability of the 'thirsty' intention at the beginning but later on drops gradually, as the 'tired' intention gradually becomes dominant (Figure 5d). Besides, the intention transition matrix learnt by HIAVI (Figure 5e) suggests that while both the 'thirsty' and 'tired' intentions are stable, the probability that the animals switch from 'thirsty' to 'tired' is relatively larger than the other way around. These observations indicate the following interpretations on the mice behavior within this cohort, which also align well with our intuition: The water-restricted mice are eager to find the water port as the trial starts, and then gradually return back to the labyrinth entrance as they obtain enough water. Once they have decided to return, the probability to go back to search for the water port would be quite low.

When comparing the dataset collected from the water-unrestricted animals, HIAVI also had a larger test log-likelihood and a lower BIC value than DIRL (Figure 6a and 6b). Interestingly, for this dataset, increasing the number of intentions seemed not to improve the performance of DIRL at all, which might be a result of a more random behavior of the animals due to the lack of motivation to find water. We then selected HIAVI with $K = 2$ according to BIC (Figure 6b) to analyze the recovered policy and intention dynamics. For this cohort of animals, the inferred policy under two intentions exhibits 'exploring' and 'tired' behavior. The policy under 'exploring' tends to encourage the animal lingering in the labyrinth, whereas the policy under

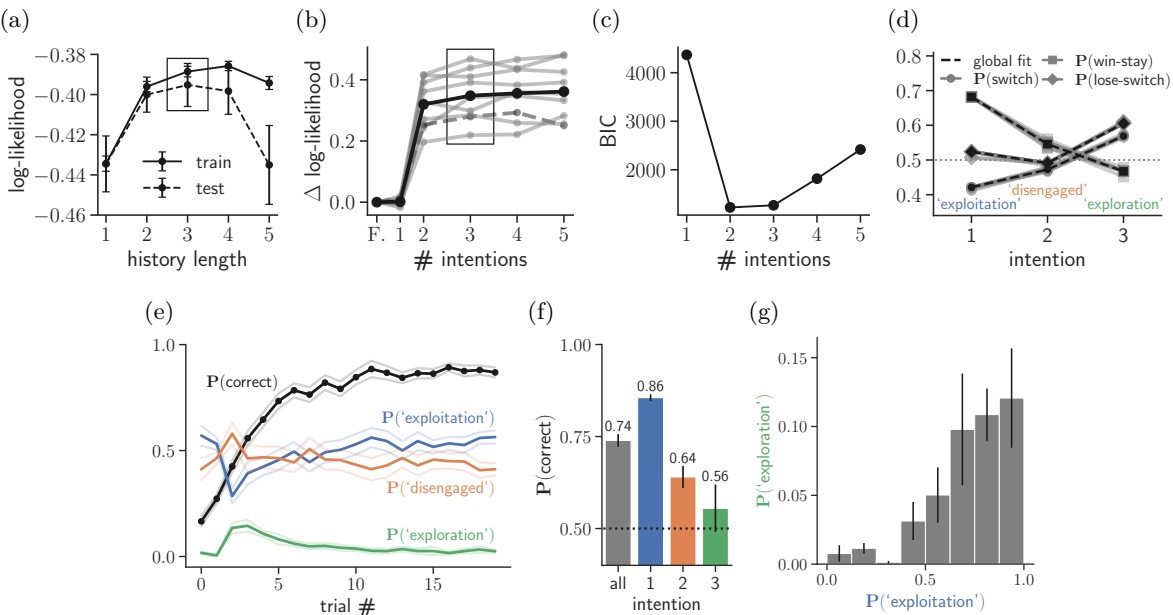

Figure 7: Results for the mice reversal-learning dataset. (a) Log-likelihood (mean ± standard error, 5-fold cross-validation) as a function of history length of single intention HIQL. (b) Change in test set log-likelihood as a function of the number of intentions in HIQL with $\ell_h = 3$, relative to the fQ-learning model (labeled 'F.'). Each trace represents a single mouse, averaged over cross-validation. Solid black indicates the mean across animals, and the dashed curve indicates the example mouse. (c) BIC as a function of the number of intention in HIQL with $\ell_h = 3$. (d) Learnt mice policy represented with the probability of switch, win-stay, and lose-switch. Each grey curve denotes one mouse. (e) Average task performance and the predicted intention dynamics. Solid and shaded curves denote the mean and standard error across 9 animals. (f) Overall task performance (gray) and the performance under different intentions, mean ± standard error across 9 animals. (g) Relationship between the probability of the 'exploitation' intention, 5 trials before block switch and the probability of the 'exploration' intention, 5 trials after block switch, mean ± standard error across 9 animals.

'tired' intention steers the animal back to the maze entrance (Figure 6c). Correspondingly, the posterior probability of 'exploring' initially dominates at the beginning of the trial but is generally surpassed by the 'tired' intention over time (Figure 6d). The inferred intention transition matrix (Figure 6e) suggests that the 'exploring' intention is unstable, meaning that it has a larger probability of switching to 'tired' intention compared to staying at 'exploring'. In contrast, after switching to the 'tired' intention, animals tend to keep this intention until the end of trial. These observations are again consistent with our intuition. Since the water-seeking motivation for this group of water-unrestricted mice is very low, they would prefer to explore the labyrinth, rather than search for the water port.

In conclusion, the above results show that HIAVI outperforms DIRL in predicting animal behavior on this labyrinth navigation benchmark for both two cohorts of mice, and is able to generate interpretable reward functions underlying different intentions. On the other hand, the significant difference between the prediction performance of HIAVI and DIRL indicates that the intention transition dynamics during natural decision-making behavior may be better described with a step-function, rather that a smoothly varying function as assumed by DIRL.

## 5.3 Application to mice reversal-learning behavior

Finally, we apply the HIQL algorithm to behavioral data recorded from a group of mice engaged in a dynamic two-armed bandit reversal-learning task. The dataset was initially reported by De La Crompe et al. (2023).

At the beginning of the task, water-restricted mice may choose from two available spouts, *left* (L) and *right* (R), with random one of them assigned water as extrinsic reward. After reaching an online performance of 75% correct in a 15-trials sliding average window and a minimum 20-trials block, the rewarded spout is changed. To formulate the MDP, we define the action space as: $\mathcal{A} = \{left, right\}$. Every state $s \in \mathcal{S}$ is defined with a tuple of truncated history information: $s_t = (\varphi_{t-1}, \ldots, \varphi_{t-\ell_h}; a_{t-1}, \ldots, a_{t-\ell_h})$, where the positive integer $\ell_h$ denotes the history length, $a \in \mathcal{A}$ denotes history action, and $\varphi \in \{correct, error\}$ represents trial feedback, *i.e.*, the extrinsic reward. Such MDP formulation allows us to avoid explicitly establishing a partially observable MDP. Different from the first two experiments, we applied the model-free variant of HIQL in this dynamic reversal-learning task since the environment model is unknown.

We begin from selecting the hyperparameter $\ell_h$ using single intention HIQL (equivalent to vanilla IQL). The dataset for all subsequent analysis consists of 64 sessions (trajectories) from 9 different animals and a 5-fold cross-validation was applied over the trajectories. We compared the log-likelihood on training and test sets of multiple IQL fitting with different $\ell_h$ (Figure 7a). The log-likelihood on test sets shows a bell-shaped curve as $\ell_h$ increases, indicating an overfit on the training set when $\ell_h > 3$. Note that there is an abnormal drop on training set log-likelihood at large $\ell_h$s. This can be explained with the insufficient sampling given the fixed set of expert demonstrations, since the size of the state space grows exponentially as the history length $\ell_h$ increases. The best test log-likelihood was achieved by $\ell_h = 3$, which was selected for subsequent steps. Next, to determine the number of intentions $K$ under which mice demonstrated the trajectories, we fit multiple HIQL with varying number of intentions. In this step, we additionally applied a forgetting Q-learning (fQ-learning) model (Beron et al., 2022), which has been widely recognized as a prominent forward behavioral model for the reversal-learning task, serving as a baseline for comparative analysis. We found that the multiple intention HIQL fitting substantially outperformed the single intention models (Figure 7b). The BIC w.r.t. different $K$ indicates that both $K = 2$ and $K = 3$ are reasonable values (Figure 7c). Taking biological interpretability into account, we selected $K = 3$ for the following discussion. We list additional details about the above procedure of fitting HIQL to this mice reversal-learning dataset in Appendix C.3.

The inferred mice policies from HIQL represent different strategies in the task under three intentions (Figure 7d). The policy under intention 1 ('exploitation') displays a strong preference of a 'win-stay' and 'lose-switch' strategy, which is the optimal policy in this deterministic reward bandit task, *i.e.*, stay on the same side for the next trial if it is rewarded in the current trial, and switch to the other one otherwise. On the other hand, under intention 2 ('disengaged'), the policy exhibits a preference for exploitation when the previous trial was successful, but following error trials, it employs a random action selection, indicated by the ca. 0.5 probability of executing a 'lose-switch'. Lastly, under intention 3 ('exploration'), the subject consistently favors selecting the option opposite to the one chosen in the preceding trial, irrespec-

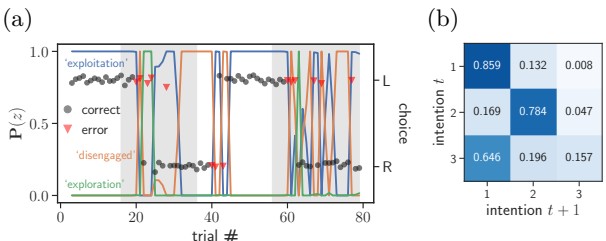

Figure 8: (a) Predicted intention dynamics for an example session. Dots and triangles indicate mice action. (b) Inferred intention transition matrix for this example mouse.

tive of whether they had won or lost in that particular instance. As shown in an example session (Figure 8a), the 'exploration' intention predominantly occurs at the onset of a block and only lasts for a few trials, whereas the other two intentions may persists over multiple consecutive trials. These results suggests that compared to 'exploitation' and 'disengaged', the 'exploration' intention is not stable, which aligns well with the learned intention transition matrix (Figure 8b). Additionally, it becomes evident that error trials tend to coincide with the trials where the posterior probability of 'disengaged' and 'exploration' intentions are dominant. The intention dynamics averaged across all blocks closely resembles those observed in the example session (Figures 7e). As each block begins with the animal's performance at a relatively low level, there is a decline in the posterior probability associated with the 'exploitation' intention, accompanied by an increase in the probabilities of the other two intentions associated with suboptimal exploratory strategies. Then as the subjects' performance steadily improves, the 'exploitation' intention progressively reasserts its dominance. In contrast to the cohort's general correct rate of $0.74 \pm 0.02$, the subjects performed significantly better

within the 'exploitation' intention, achieving a correctness rate of $0.86 \pm 0.01$, whereas they only attained lower correctness rates of $0.64 \pm 0.03$ and $0.56 \pm 0.06$ in the two other intentions (Figure 7f). Interestingly, it was observed that the mean posterior probability of the 'exploration' intention at the beginning of a new block showed a positive correlation with the average probability of the 'exploitation' state at the end of the preceding block (Figure 7g). These results confirm the difference between the two suboptimal intentions 'disengaged' and 'exploration': the first strategy explores passively upon error, while the second strategy involves a deliberate, exploration-oriented action selection when the subject is highly engaged and possess a good understanding of the environment. Although the 'exploration' intention is suggested to be unstable, it provides the opportunity to characterize a type of learning strategy based on error, which has been shown experimentally by Kononowicz et al. (2022).

In summary, by applying HIQL in a real-world mice reversal learning dataset, we demonstrate the possibility of performing model-free learning and producing interpretable behavior characteristics of animal experts via HIQL. We also detected a typical exploration strategy of mice during this value-based decision-making task via mathematical modeling, which has only be achieved in perceptual decision-making tasks (Ashwood et al., 2022b). Last but not the least, the model training procedure that we proposed in this experiment enables fits to individual subjects without running into the challenge of post hoc alignment of intentions from different individuals. Although the mice subjects exhibited similar behavior during our experiment, limiting the magnitude of observed differences in individual solutions, we anticipate that this method will prove beneficial in tasks where subjects exhibit greater variance in policy preferences.

## 6 Conclusion

This paper proposes the class of EM-based *hierarchical inverse Q-learning* (HIQL) algorithms, extending the IQL algorithm (Kalweit et al., 2020) for multi-intention IRL problems. Empirical results demonstrate that HIQL outperforms the state-of-the-art on both synthesized and real-world datasets, and is able to produce interpretable behavior characteristics. We also mathematically characterized typical exploration behavior of rodents during value-based decision-making using HIQL.

Compared to the state-of-the-art algorithm for characterizing animal behavior, DIRL (Ashwood et al., 2022a), the advantages of our HIQL algorithm are two fold. First, the assumptions about the underlying intention transition dynamics in HIQL align better with those observed in real-world behavioral experiments that animal and human alternate between discrete strategies during decision making (Ashwood et al., 2022b), rather than optimize their policy over a continuous time-varying reward as assumed by DIRL. Second, the inner-loop IRL problem solver adapted by HIQL, the class of IQL algorithms, were shown to have better performance in learning the expert's unknown reward and higher computing efficiency than the maximum entropy IRL algorithm used by DIRL (Kalweit et al., 2020).

As future work, we plan to relax the Markov assumption about the intention transition dynamics in HIQL to incorporate non-Markovian intention dynamics, and also extend HIQL for an unknown number of reward functions (Dimitrakakis & Rothkopf, 2012; Michini & How, 2012; Surana & Srivastava, 2014).

### Acknowledgments

This work has been funded as part of BrainLinks-BrainTools, which is funded by the Federal Ministry of Economics, Science and Arts of Baden-Württemberg within the sustainability programme for projects of the Excellence Initiative II; as well as the Bernstein Award 2012, the Research Unit 5159 "Resolving Prefrontal Flexibility" (Grant DI 1908/11-1), and the Deutsche Forschungsgemeinschaft (Grants DI 1908/3-1 and DI 1908/6-1) to I.D., and CRC/TRR 384 "IN-Code" to I.D. and J.B.

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

# A   Theoretical and technical details

## A.1   Proof of Theorem 4.3

*Proof.* The objective function $J\left(\Theta^+ \mid \Theta\right)$ from problem (6) can be written as:

$$
J\left(\Theta^+ \mid \Theta\right) = \underset{\xi \sim \mathcal{D}, \eta}{\mathbf{E}}\left[\log \mathbf{P}\left(\xi, \eta \mid \Theta^+\right)\right]
$$

$$
= \underset{\xi \sim \mathcal{D}}{\mathbf{E}}\left[\sum_\eta \mathbf{P}(\eta \mid \xi, \Theta)\log \mathbf{P}\left(\xi, \eta \mid, \Theta^+\right)\right]
$$

$$
= \underset{\xi \sim \mathcal{D}}{\mathbf{E}}\left[\sum_\eta \mathbf{P}(\eta \mid \xi, \Theta)\log\left(\Pi_{z_0}^+ \pi_{r_{z_0}^+}(s_0, a_0)\prod_{t=1}^n \Lambda_{z_{t-1}z_t}^+ \pi_{r_{z_t}^+}(s_t, a_t)\right)\right]
$$

$$
= \underset{\xi \sim \mathcal{D}}{\mathbf{E}}\left[\sum_\eta \mathbf{P}(\eta \mid \xi, \Theta)\log \Pi_{z_0}^+\right] + \underset{\xi \sim \mathcal{D}}{\mathbf{E}}\left[\sum_\eta \mathbf{P}(\eta \mid \xi, \Theta)\sum_{t=1}^n \log \Lambda_{z_{t-1}z_t}^+\right]
$$

$$
+ \underset{\xi \sim \mathcal{D}}{\mathbf{E}}\left[\sum_\eta \mathbf{P}(\eta \mid \xi, \Theta)\sum_{t=0}^n \log \pi_{r_{z_t}^+}(s_t, a_t)\right]
$$

$$
= \underset{\xi \sim \mathcal{D}}{\mathbf{E}}\left[\sum_{i=1}^K \underbrace{\sum_\eta \mathbf{P}(\eta \mid \xi, \Theta) I_i(z_0)}_{\mathbf{P}(z_0 = i \mid \xi, \Theta)}\log \Pi_i^+\right] + \underset{\xi \sim \mathcal{D}}{\mathbf{E}}\left[\sum_{i=1}^K \sum_{j=1}^K \sum_{t=1}^n \underbrace{\sum_\eta \mathbf{P}(\eta \mid \xi, \Theta) I_i(z_{t-1})I_j(z_t)}_{\mathbf{P}(z_{t-1}=i, z_t = j \mid \xi, \Theta)}\log \Lambda_{ij}^+\right]
$$

$$
+ \underset{\xi \sim \mathcal{D}}{\mathbf{E}}\left[\sum_{i=1}^K \sum_{t=0}^n \underbrace{\sum_\eta \mathbf{P}(\eta \mid \xi, \Theta) I_i(z_t)}_{\mathbf{P}(z_t = i \mid \xi, \Theta)}\log \pi_{r_i^+}(s_t, a_t)\right]
$$

$$
= \underset{\xi \sim \mathcal{D}}{\mathbf{E}}\left[\sum_{i=1}^K \mathbf{P}(z_0 = i \mid \xi, \Theta)\log \Pi_i^+\right] \tag{A.1a}
$$

$$
+ \underset{\xi \sim \mathcal{D}}{\mathbf{E}}\left[\sum_{i=1}^K \sum_{j=1}^K \sum_{t=1}^n \mathbf{P}(z_{t-1} = i, z_t = j \mid \xi, \Theta)\log \Lambda_{ij}^+\right] \tag{A.1b}
$$

$$
+ \underset{\xi \sim \mathcal{D}}{\mathbf{E}}\left[\sum_{i=1}^K \sum_{t=0}^n \mathbf{P}(z_t = i \mid \xi, \Theta)\log \pi_{r_i^+}(s_t, a_t)\right], \tag{A.1c}
$$

where $I_i$ is the indicator function with $I_i(x) = 1$ for all $x = i$ and 0 otherwise. Thus maximizing $J\left(\Theta^+ \mid \Theta\right)$ over $\Theta^+$ (problem (6)) is equivalent to separately maximizing (A.1a) over $\Pi^+$, (A.1b) over $\Lambda^+$, and (A.1c) over $\mathcal{R}^+ = \left\{r_1^+, \ldots, r_K^+\right\}$. The first two optimization problems can be formally written as:

$$
\begin{aligned}
&\text{maximize (over } \Pi^+) && \mathbf{E}_{\xi \sim \mathcal{D}}\left[\sum_{i=1}^K \mathbf{P}(z_0 = i \mid \xi, \Theta)\log \Pi_i^+\right]\\
&\text{subject to} && \Pi^+ \succeq 0,\ \mathbf{1}^T \Pi^+ = 1,
\end{aligned} \tag{7}
$$

and

$$
\begin{aligned}
&\text{maximize (over } \Lambda^+) && \mathbf{E}_{\xi \sim \mathcal{D}}\left[\sum_{i=1}^K \sum_{j=1}^K \sum_{t=1}^n \mathbf{P}(z_{t-1} = i, z_t = j \mid \xi, \Theta)\log \Lambda_{ij}^+\right]\\
&\text{subject to} && \Lambda_{i:}^+ \succeq 0,\ \mathbf{1}^T \Lambda_{i:}^+ = 1,\quad i = 1, \ldots, K.
\end{aligned} \tag{8}
$$

Note that (A.1c) has similar structure as the optimization objective of IRL (problem (1)), and can be further separated for each $r \in \mathcal{R}$, thus maximizing (A.1c) over $\mathcal{R}^+$ is equivalent to independently solving an IRL

problem for each $r_i$, with the $t$th demonstration weighted by $\mathbf{P}(z_t = i \mid \xi, \Theta)$:

$$
\begin{aligned}
&\text{maximize (over } r_i^+) \quad \mathbf{E}_{\xi \sim \mathcal{D}} \left[ \sum_{t=0}^n \mathbf{P}(z_t = i \mid \xi, \Theta) \log \pi_{r_i^+}(s_t, a_t) \right] \\
&\text{subject to} \quad \pi_{r_i^+}(s, a) = \exp\left( Q(s, a) - \log \sum \exp Q(s, \cdot) \right), \text{ for all } s \in \mathcal{S}, \, a \in \mathcal{A} \\
&\qquad Q(s, a) = r_i^+(s, a) + \gamma \sum_{s' \in \mathcal{S}} P(s, a, s') \max_{a' \in \mathcal{A}} Q(s', a'), \text{ for all } s \in \mathcal{S}, \, a \in \mathcal{A}.
\end{aligned} \tag{9}
$$

Similar to (1), the constraints on policy $\pi_{r_i^+}$ are introduced here to make the IRL problem tractable. □

### A.2 Computing required posterior probabilities

To obtain the posterior probability terms $\mathbf{P}(z_0 = i \mid \xi, \Theta)$, $\mathbf{P}(z_{t-1} = i, z_t = j \mid \xi, \Theta)$, and $\mathbf{P}(z_t = i \mid \xi, \Theta)$ required for evaluating the objective functions of (7), (8) and (9), we apply the Baum-Welch algorithm (Baum & Petrie, 1966) as follows.

Let $\alpha_t \in \mathbf{R}^K$ be the forward probability which denotes the posterior probability of observing the expert demonstrations up until time $t$ and the $t$th demonstration was generated under reward function $r_i$, i.e.,

$$
\alpha_t(i) = \mathbf{P}(\xi_{0:t}, z_t = i \mid \Theta) \tag{A.2}
$$

$$
= \begin{cases} \Pi_i \pi_{r_i}(s_0, a_0) & t = 0 \\ \alpha_{t-1}^T \Lambda_{:i} \pi_{r_i}(s_t, a_t) & t > 0, \end{cases} \tag{A.3}
$$

for all $i = 1, \ldots, K$. Similarly, we define the backward probability $\beta_t \in \mathbf{R}^K$ denoting the posterior probability of observing the expert demonstrations after time $t$, given that the $t$th demonstration was generated under reward function $r_i$:

$$
\beta_t(i) = \mathbf{P}(\xi_{t+1:n} \mid z_t = i, \Theta) \tag{A.4}
$$

$$
= \begin{cases} \sum_{j=1}^K \beta_{t+1}(j) \Lambda_{ij} \pi_{r_j}(s_{t+1}, a_{t+1}) & t < n \\ 1 & t = n, \end{cases} \tag{A.5}
$$

for all $i = 1, \ldots, K$. With (A.3) and (A.5), we can efficiently obtain the forward and backward probabilities for each demonstration in a recursive form. Finally, we can compute the posterior probabilities

$$
\mathbf{P}(z_t = i \mid \xi, \Theta) = \frac{\alpha_t(i) \beta_t(i)}{\alpha_t^T \beta_t}, \tag{A.6}
$$

and

$$
\mathbf{P}(z_{t-1} = i, z_t = j \mid \xi, \Theta) = \frac{\alpha_t(i) \Lambda_{ij} \pi_{r_j}(s_{t+1}, a_{t+1}) \beta_{t+1}(j)}{\sum_{i'=1}^K \sum_{j'=1}^K \alpha_t(i') \Lambda_{i'j'} \pi_{r_{j'}}(s_{t+1}, a_{t+1}) \beta_{t+1}(j')}. \tag{A.7}
$$

# B Algorithms

---

**Algorithm 1** Inverse action-value iteration.

---

**given** expert demonstrations $\mathcal{D}$.

1: **initialize** $r$, $Q$.
2: Obtain expert policy $\pi^{\mathrm{E}}$ by counting the state-action visitation in $\mathcal{D}$.
3: **repeat**
4:     **for all** $s \in \mathcal{S}$ **do**
5:         Calculate $\mu(s,a) \coloneqq \log(\pi^{\mathrm{E}}(s,a)) - \gamma \sum_{s' \in \mathcal{S}} P(s,a,s') \max_{a' \in \mathcal{A}} Q(s',a')$ for all $a \in \mathcal{A}$.
6:         Update $r(s,\cdot)$ by solving (5) via least squares.
7:         Update $Q(s,a) \coloneqq r(s,a) + \gamma \sum_{s' \in \mathcal{S}} P(s,a,s') \max_{a' \in \mathcal{A}} Q(s',a')$ for all $a \in \mathcal{A}$.
8:     **end for**
9: **until** stopping criterion is satisfied.

---

**Algorithm 2** Inverse Q-learning.

---

**given** expert demonstrations $\mathcal{D}$; learning rate $\alpha_r$, $\alpha_Q$, and $\alpha_{\mathrm{Sh}}$.

1: **initialize** $r$, $Q$, $Q^{\mathrm{Sh}}$, and state-action visitation counter $\rho$.
2: **repeat**
3:     **for all** $\xi \in \mathcal{D}$ **do**
4:         **for** $t = 0, \ldots, n-1$ **do**
5:             Observe state-action pairs $(s_t, a_t, s_{t+1})$ from $\xi$, and update $\rho(s_t, a_t) \coloneqq \rho(s_t, a_t) + 1$.
6:             Obtain $\pi^{\mathrm{E}}(s_t, a) \coloneqq \rho(s_t, a) / \sum_{a' \in \mathcal{A}} \rho(s_t, a')$ for all $a \in \mathcal{A}$.
7:             Update $Q^{\mathrm{Sh}}(s_t, a_t) \coloneqq (1 - \alpha_{\mathrm{Sh}})Q^{\mathrm{Sh}}(s_t, a_t) + \alpha_{\mathrm{Sh}} \cdot \gamma \max_{a \in \mathcal{A}} Q(s_{t+1}, a)$.
8:             Calculate $\mu(s_t, a) \coloneqq \log(\pi^{\mathrm{E}}(s_t, a)) - Q^{\mathrm{Sh}}(s_t, a)$ for all $a \in \mathcal{A}$.
9:             Update $r(s_t, a_t) \coloneqq (1 - \alpha_r)r(s_t, a_t) + \alpha_r \left( \mu(s_t, a_t) + \frac{1}{\mathbf{card}(\mathcal{A}_{-a_t})} \sum_{b \in \mathcal{A}_{-a_t}} (r(s_t, b) - \mu(s_t, b)) \right)$.
10:           Update $Q(s_t, a_t) \coloneqq (1 - \alpha_Q)Q(s_t, a_t) + \alpha_Q(r(s_t, a_t) + \gamma \max_{a \in \mathcal{A}} Q(s_{t+1}, a))$.
11:         **end for**
12:     **end for**
13: **until** stopping criterion is satisfied.

---

**Algorithm 3** Hierarchical inverse Q-learning.

---

**given** expert demonstrations $\mathcal{D}$ and reward set cardinality $K$.

1: **initialize** $\Pi$, $\Lambda$, $r_1, \ldots, r_K$.
2: **repeat**
3:     Calculating the Boltzmann policy $\pi_{r_i}$ for all $i = 1, \ldots, K$.
4:     **for all** $\xi \in \mathcal{D}$ **do**
5:         **for** $i = 1, \ldots, K$, $j = 1, \ldots, K$, $t = 0, \ldots, n$ **do**
6:             Calculate $\alpha_t(i)$ and $\beta_t(i)$ according to (A.3) and (A.5).
7:             Calculate $\mathbf{P}(z_t = i \mid \xi, \Theta)$ and $\mathbf{P}(z_{t-1} = i, z_t = j \mid \xi, \Theta)$ according to (A.6) and (A.7).
8:         **end for**
9:     **end for**
10:     Update $\Pi$ and $\Lambda$ according to (10) and (11).
11:     **for** $i = 1, \ldots, K$ **do**
12:         **initialize** expert demonstration subset $\mathcal{D}'$.
13:         **for all** $\xi \in \mathcal{D}$ **do**
14:             **for** $t = 0, \ldots, n-1$ **do**
15:                 Observe state-action pairs $(s_t, a_t, s_{t+1})$ from $\xi$; sample $u \sim \mathcal{U}(0,1)$.
16:                 **if** $u < \mathbf{P}(z_t = i \mid \xi, \Theta)$ **then**
17:                     $\mathcal{D}' \coloneqq \mathcal{D}' \cup \{(s_t, a_t, s_{t+1})\}$.
18:                 **end if**
19:             **end for**
20:         **end for**
21:         Update $r_i$ by applying Algorithm 1 to $\mathcal{D}'$ **if** $P$ is known, **else** using Algorithm 2.
22:     **end for**
23: **until** stopping criterion is satisfied.

---

## C Datasets and model training

### C.1 Gridworld benchmark

The following pseudo-code shows the expert policy for each episode that we used to obtain expert demonstrations in the gridworld environment in §5.1.

---

1:  **initialize** $s := (0,0)$, $\pi := \pi^{\text{goal}}$, $t := 0$.
2:  **repeat**
3:      $a \sim \pi$.
4:      $s \sim P(s, a, \cdot)$.
5:      **if** $s$ has barrier '#' **then**
6:          Switch to another policy (30%).
7:      **else if** $t = 8$ **then**
8:          $\pi := \pi^{\text{abandon}}$ (50%).
9:      **end if**
10:     $t := t + 1$.
11: **until** $(0,0)$ or $(4,4)$ is reached.

---

The whole expert demonstration dataset consisted of 1024 trajectories. All evaluated algorithms were fit to multiple sub-datasets with different number of expert trajectories, and each dataset was analyzed using a 5-fold cross-validation. Note that since the EM process does not guarantee to find the global optimum, for each training, HIAVI was randomly initialized repeatedly for 10 times and the best performed initialization (evaluated by test set log-likelihood) was selected for analysis. The initial intention distribution $\Pi$ was initialized uniformly, and the intention transition matrix $\Lambda$ was initialized as: $\Lambda = 0.95 \times I + \mathcal{N}(0, 0.05 \times I)$, where $\mathcal{N}$ denotes the normal distribution and $I \in \mathbf{R}^{2 \times 2}$ is the identity matrix. This initial $\Lambda$ was then normalized so that each row added up to 1. The discount factor of the MDP was set to be $\gamma = 0.9$.

### C.2 Real-world mice navigation benchmark

For comparability with the results in Ashwood et al. (2022a), we obtained their pre-processed mouse trajectories for water-restricted and water-unrestricted animals from `https://github.com/97aditi/dynamic_irl`. The original recorded animal trajectories from Rosenberg et al. (2021) are provided with MIT open source license at the following repository: `https://github.com/markusmeister/Rosenberg-2021-Repository`. For the pre-processing, Ashwood et al. (2022a) used a clustering algorithm (based on DBSCAN (Ester et al., 1996)) for aligning trajectories across animals and bouts to reduce variability. After the pre-processing, they obtained 200 trajectories from the water-restricted animals and 207 trajectories from the water-unrestricted animals. 20% of trajectories from each cohort were held out as a test set.

We used the source code and the best performed set of hyperparameters provided by Ashwood et al. (2022a) to train DIRL on the animal trajectories. All HIAVI algorithms were trained for 10 repeated runs with different initializations, and the results from the initializations with the highest test set log-likelihood was selected for analysis. The initial intention distribution $\Pi$ was initialized uniformly, and the intention transition matrix $\Lambda$ was initialized as: $\Lambda = 0.95 \times I + \mathcal{N}(0, 0.05 \times I)$, where $\mathcal{N}$ denotes the normal distribution and $I \in \mathbf{R}^{K \times K}$ is the identity matrix. This initial $\Lambda$ was then normalized so that each row added up to 1. The MDP was defined to have a deterministic state transition function $P$ and the discount factor was set to be $\gamma = 0.99$.

### C.3 Application to mice reversal-learning behavior

The expert demonstrations for this dynamic reversal-learning task was collected from a cohort of mice consisted of 9 mice in total. Behavior recordings for each subject were repeated for at least 7 independent sessions with an average of ca. 87 trials per session.

We employed a multi-stage fitting procedure to select hyperparameters and to allow us to fit HIQL individually to each animal. In the first stage, we concatenated the data from all animals in a single dataset

together. We then performed multiple IQL (single intention HIQL) with different history length $\ell_h = 1, \ldots, 5$ on the concatenated dataset. Out of the 5 different values, We chose the $\ell_h$ that resulted in the best test set log-likelihood for subsequent stages. In the second stage, we run multiple HIQL with different number of intentions $K = 2, \ldots, 5$ again to the concatenated dataset to obtain a global fit. For each fitting, we performed 10 different initializations and the best performed initialization was selected. The initial intention distribution $\Pi$ was initialized uniformly, and the intention transition matrix $\Lambda$ was initialized as: $\Lambda = 0.95 \times I + \mathcal{N}(0, 0.05 \times I)$, where $\mathcal{N}$ denotes the normal distribution and $I \in \mathbf{R}^{K \times K}$ is the identity matrix. This initial $\Lambda$ was then normalized so that each row added up to 1. In the last stage of the fitting procedure, we wanted to obtain an independent but aligned HIQL fit for each animal, so we initialized the parameters for each animal with the best global fit parameters from all animals together, omitting the necessity to permute the retrieved intentions from each animal so as to map semantically similar intentions to one another. A 5-fold cross-validation was used to split the training and test dataset. The discount factor fot this experiment were set to be $\gamma = 0.99$.

## D   Code availability

Implementation for the class of HIQL algorithms can be found at `https://github.com/haozhu10015/hiql`.

