# OpenReview forum: "Multi-intention Inverse Q-learning for Interpretable Behavior Representation"
_TMLR — Accepted by TMLR_

### Review · Reviewer_7Eh8 · 2024-07-07

**Summary Of Contributions:**

The authors propose a new class of algorithms, Hierarchical Inverse Q-learning (HIQL), including a model-based (HIAVI) and a model-free (HIQL) algorithm. The general goal is to accurately determine the underlying reward function of a set of expert demonstrations. The authors suggest that reward functions (intentions) during one trajectory change according to a Markov Chain. HIQL approximates the initial state distribution and state transition matrix of the Markov Chain using Expectation-Maximisation on the expert trajectories. It then selects the most likely policy under the state and transition distribution using Inverse Q-Learning. The authors provide theoretical results (a derivation of the objective function) and empirical results (evaluating HIAVI on a toy example and real-world mice data, and HIQL on a real-world mice bandit problem).

HIQL is different from previous methods as it assumes that intentions change discretely within a trajectory, in contrast to previous methods, which typically assumed continuous transitions between intentions. The authors show that HIQL performs distinctively different and better than previous methods, such as Dynamic Inverse Reinforcement Learning (DIRL).

**Audience:**

Yes

**Broader Impact Concerns:**

The reviewer does not see any ethical concerns with the method proposed. The authors could include an ethics section in the appendix, describing the potential ethical concerns of inferring intentions of human behaviour.

**Claims And Evidence:**

Yes

**Requested Changes:**

Overall, it is the reviewers opinion that the paper is sufficient in quality and further improvements will not affect the decision substantially. If the authors could address the questions in the Weaknesses section, the clarity of the paper could be further improved.

Additionally, more complex experiments with many intentions and high-dimensional state spaces would obviously improve the potential impact of the paper drastically. However, the results are sufficient to support the claims in the paper and provide interesting insights for the TMLR community.

**Strengths And Weaknesses:**

Strengths:
- The paper is well written and fairly easy to follow. The notation is not overbearing and the algorithms are clearly explained.
- HIQL is well placed into related work and the difference to previous methods is clear.
- The empirical results are analysed extensively and all design decisions are motivated well across all three experiments.


Weaknesses:
- The language around "demonstrations", "trajectories", "reward functions/maps", and "intentions" could be clearer in the text. Figure 1 clarifies that a "demonstration" is one state transition within an expert trajectory. However, it was just slightly confusing from the text in Section 4, if a demonstration was a full trajectory or one transition. For example "Inverse Q-learning. Given the set of expert demonstrations $\mathcal{D}$ in an MDP, where each trajectory $\xi \in \mathcal{D}$ is denoted with a sequence of state-action pairs: $\xi=\{(s_1, a_1), \ldots,(s_n, a_n)\}$, the IRL problem consists in determining a reward function $r$ that best explains the observed expert behavior in the form of demonstrated trajectories.". One can infer that a demonstration is a state transition from the definition of $\eta=\{z_1, \ldots, z_n\}$ but the paper could be improved by just quickly defining that a demonstration is a state transition.
- Similarly, the almost interchangeable use of "reward", "intention", and "reward map" makes the paper slightly harder to understand, as "intention" is not clearly defined. For example, why did the authors not reuse the word "reward map" as used in DIRL? Should "intention" always be used when reward functions change discretely and "map" when the reward changes continuously?
- Figure 3c (1 intention) could be explained better. “Since the single-intention variants assume all trajectories were demonstrated under one intention, the EVD was analyzed twice on the ground truth reward for different intentions with the same learnt Boltzmann policy“. Does that mean that the authors ran HIAVI with 1 Intention multiple times, and hand-selected the resulting policies most akin to 'goal' and 'abandon'? Or how is it possible that you run HIAVI "for different intentions", when the intentions are learned unsupervised? Is HIAVI more likely to learn 'goal' or 'abandon' if the intention=1?
- While the authors provide information about the mice dataset in the appendix, the reviewers preference is that specific information, such as the number of expert trajectories would be present in the main body of the paper. For example, when the authors report standard error, it is important that it's clearly stated if the standard error is reported over how many trajectories (and potentially over different seeds for initial state and state transition distribution?)
- The experiments do not appear particularly complex, compared to other IRL problem settings as mentioned in the Introduction (self-driving, robotics, etc.). While the reviewer does not deem it necessary, obviously the paper could be made much stronger by demonstrating HIQL on datasets with 10s or 100s of intentions with much larger state spaces.

---

> ### Author Response · Authors · 2024-07-31
>
> We thank reviewer 7Eh8 for the questions and suggestions to improve our submission. We address your concerns in detail below.
>
> `"The language around "demonstrations", "trajectories", "reward functions/maps", and "intentions" could be clearer in the text. Figure 1 clarifies that a "demonstration" is one state transition within an expert trajectory. However, it was just slightly confusing from the text in Section 4, if a demonstration was a full trajectory or one transition."`
>
> `“While the authors provide information about the mice dataset in the appendix, the reviewers preference is that specific information, such as the number of expert trajectories would be present in the main body of the paper. For example, when the authors report standard error, it is important that it's clearly stated if the standard error is reported over how many trajectories (and potentially over different seeds for initial state and state transition distribution?)”`
>
> Thanks for your suggestion, we have updated the manuscript according to your comments and added corresponding information in the latest revision.
>
> `“Similarly, the almost interchangeable use of "reward", "intention", and "reward map" makes the paper slightly harder to understand, as "intention" is not clearly defined. For example, why did the authors not reuse the word "reward map" as used in DIRL? Should "intention" always be used when reward functions change discretely and "map" when the reward changes continuously?”`
>
> The word ‘intention’ was widely used in literature about IRL problems with non-stationary reward functions (cf. the references listed in section 2), and it is commonly assumed that each intention corresponds to one specific reward function. Thanks for your suggestion, we have made this point clearer in the latest revision. The phrase ‘reward map’ was initially introduced in the DIRL paper (Ashwood et al. NeurIPS 2022). The authors from DIRL might have multiple reasons to introduce the new phrase ‘reward map’ instead of reusing the word ‘intention’, but we think the widely used definition of ‘intention’ would fit better to our description in this work. However, we decided to keep using the phrase ‘reward map’ in this paper when talking about DIRL in order that the readers do not have to translate the word back and forth when comparing our paper with the DIRL paper.

---

> > ### Author Response · Authors · 2024-07-31
> >
> > `“The experiments do not appear particularly complex, compared to other IRL problem settings as mentioned in the Introduction (self-driving, robotics, etc.). While the reviewer does not deem it necessary, obviously the paper could be made much stronger by demonstrating HIQL on datasets with 10s or 100s of intentions with much larger state spaces.”`
> >
> > The proposed HIQL algorithm in this work aims at providing an interpretable representation about animal and human behavior, where the number of intentions that an animal subject may have would be in general limited (Ashwood et al. Nat. Neuroscience (2022)). On the other hand, our method does not have a constraint on the total number of intentions that the expert might have during its demonstration, meaning that in theory HIQL is able to be applied to environments where the expert has unlimited number of intentions (with the cost of running time). Nevertheless, we propose that it would be more valuable and have better interpretability to learn a hierarchical intention model under more complex environments, instead of just simply increasing the number of intentions, which is above the claims in this paper but we believe will provide meaningful directions for future work.
> >
> > Does this address your concerns?

---

> > > ### Comment · Reviewer_7Eh8 · 2024-08-05
> > > **Thank you for the replies**
> > >
> > > I acknowledge the replies.
> > >
> > > Could the authors also address my question about Figure 3c? Thanks in advance!

---

> > > > ### Author Response · Authors · 2024-08-05
> > > >
> > > > Sorry for the late response about Figure 3c, we accidentally missed this part when copying our responses to OpenReview because of the characters limitation. Your question about Figure 3c is addressed as follows.
> > > >
> > > > `“Figure 3c (1 intention) could be explained better. “Since the single-intention variants assume all trajectories were demonstrated under one intention, the EVD was analyzed twice on the ground truth reward for different intentions with the same learnt Boltzmann policy“. Does that mean that the authors ran HIAVI with 1 Intention multiple times, and hand-selected the resulting policies most akin to 'goal' and 'abandon'? Or how is it possible that you run HIAVI "for different intentions", when the intentions are learned unsupervised? Is HIAVI more likely to learn 'goal' or 'abandon' if the intention=1?”`
> > > >
> > > > Under the single-intention setup, we ran HIAVI (which reduces to IAVI) and DIRL (which reduces to MaxEnt IRL) only once (for each fold of cross validation) on the training set, and got an output of only one learnt reward function and its corresponding learnt policy. Then the same output learnt policy was evaluated **twice** repeatedly under the different true reward functions respectively in order to provide the two state-values in Figure 3c. This means that under the single-intention case, we did not explicitly consider the ‘intentions’, but just solve the problem in the traditional single-intention IRL setup. Then evaluating the single learnt policy under different ground truth rewards repeatedly provides us the information about whether the learnt policy is more close to the ‘goal’ or the ‘abandon’ intention. The results show that the learnt policy from single-intention HIAVI exhibits a mixed characteristic of both 'goal' and 'thirsty' intentions but is more close to the former, whereas the learnt policy from single-intention DIRL deviates from both of the two intentions. This explains why HIAVI already had a better prediction log-likelihood than DIRL under the single-intention setup. We have made corresponding clarification in the current revision.

---

> > > > > ### Comment · Reviewer_7Eh8 · 2024-08-05
> > > > >
> > > > > Thank you! That addresses all my concerns. I will take the replies into account for the recommendation.

---

### Review · Reviewer_CkFD · 2024-07-07

**Summary Of Contributions:**

This paper presents *hierarchical inverse Q-learning* (HIQL), a novel algorithm for inverse reinforcement learning that assumes the reward function is non-stationary, and varies according to a discrete-time Markov process.  HIQL extends the IAVI algorithm (which estimates stationary reward functions from expert trajectories), to the setting where reward functions change at discrete points in time.  The primary motivation for the work appears to be the analysis of the behavior of human and animal subjects, rather than imitation learning.  In addition to experiments with synthetic data, they demonstrate the effectiveness of HIQL in behavioral analysis by applying it to two datasets collected from real-world experiments with mice completing navigation tasks.  Qualitative and quantitative results suggest that the reward models learned with HIQL better fit the observed behavior, and in the synthetic case, better capture the underlying dynamics of the reward function.

**Audience:**

Yes

**Broader Impact Concerns:**

I identified no future ethical concerns arising from this work that would need to be addressed in the text.

**Claims And Evidence:**

Yes

**Requested Changes:**

1. **Critical** - The paper really describes two algorithms, HIAVI and HIQL, for the cases where the transition dynamics are known and unknown respectively.  The distinction between these algorithms, and their connection to the corresponding algorithms of Kalweit et al. (2020) should be made explicit in sections 3 and 4.
2. More detailed pseudocode for HIQL and HIAVI would helpful in understanding the algorithms, as Algorithm 1 is somewhat informal in its description of each step. (These could be included in the appendices.)
3. As HIQL is a form of expectation maximization, it could be made more explicit that the reward function sequences $\xi$ are the latent parameters that are to be marginalized over.
4. It was not immediately clear in sections 5.2 and 5.3 whether the mouse data sets were collected specifically for this study, or were used in previous studies.  While this ambiguity may have been needed to maintain anonymity, any previous published uses of the data should be cited.
5. While the log-likelihood comparisons in figures 5a and 6a suggest that the Markov chain model is a better fit the the mouse dataset, it is unclear whether this simply reflects the larger number of parameters in the models learned by HIQL/HIAVI.  Comparing the BIC values for these two model space would be useful.  Section 5.3 already reports BIC values for HIQL with different numbers of parameters, so this should be straightforward.
6. As IAVI is a relatively new algorithm, a more detailed description of the method in Section 3 would be helpful.

**Strengths And Weaknesses:**

**Strengths:**

1. The HIQL algorithm itself, which makes clever use of the recently developed IAVI algorithm to allow for efficient inference of latent, non-stationary reward processes.
2. The extensive experimental analysis using both synthetic data and data from real behavioral science experiments reflecting the primary use-case for this algorithm.

**Weaknesses:**

1. The technical presentation of HIQL could be significantly improved.
2. It is not immediately clear whether the improved log-likelihood of the reward processes learned with HIQL over DIRL are due to a better fit with underlying decision making process that controls the behavior of the mice, or is simply the result of HIQL having more parameters than DIRL allowing for a tighter fit.

---

> ### Author Response · Authors · 2024-07-31
>
> We thank reviewer CkFD for the questions and suggestions to improve our submission. We address your concerns in detail below.
>
> `“Critical - The paper really describes two algorithms, HIAVI and HIQL, for the cases where the transition dynamics are known and unknown respectively. The distinction between these algorithms, and their connection to the corresponding algorithms of Kalweit et al. (2020) should be made explicit in sections 3 and 4.”`
>
> `“More detailed pseudocode for HIQL and HIAVI would helpful in understanding the algorithms, as Algorithm 1 is somewhat informal in its description of each step. (These could be included in the appendices.)”`
>
> `“As HIQL is a form of expectation maximization, it could be made more explicit that the reward function sequences are the latent parameters that are to be marginalized over.”`
>
> `“As IAVI is a relatively new algorithm, a more detailed description of the method in Section 3 would be helpful.”`
>
> `“It was not immediately clear in sections 5.2 and 5.3 whether the mouse data sets were collected specifically for this study, or were used in previous studies. While this ambiguity may have been needed to maintain anonymity, any previous published uses of the data should be cited.”`
>
> Thank you for your suggestion. We have updated the current revision according to your comments. Specifically, in section 3, we now provide a more detailed description about the class of IQL algorithms as well as corresponding algorithms (Appendix B) as the background. We also include more technical details about the adaptation from the vanilla IQL to HIQL in section 4, and the pseudocode is also updated accordingly (which is now moved to appendix for space reasons).
>
> `“While the log-likelihood comparisons in figures 5a and 6a suggest that the Markov chain model is a better fit the the mouse dataset, it is unclear whether this simply reflects the larger number of parameters in the models learned by HIQL/HIAVI. Comparing the BIC values for these two model space would be useful. Section 5.3 already reports BIC values for HIQL with different numbers of parameters, so this should be straightforward.”`
>
> Thanks for your suggestion, we have now included the BIC values for DIRL in Figure 5b and 6b, which indicates that the better performance of HIAVI was not due to a larger number of parameters.
>
> Does this address your concerns?

---

> > ### Comment · Reviewer_CkFD · 2024-08-09
> > **Response to Authors**
> >
> > I believe the changes have addressed my remaining concerns.  Thank you

---

### Review · Reviewer_NGQK · 2024-07-26

**Summary Of Contributions:**

This paper presents a method to infer discrete time-varying rewards with IRL called HIQL. Unlike previous IRL methods that only aim to infer a single reward function, HIQL is motivated by the observation that animals' intentions of tasks over time often change due to changes in their physical conditions. The authors mainly compare HIQL with one previous method, DIRL, which models the reward function as a smoothly time-varying linear combination of reward maps. In the experiments, they first demonstrate HIQL's superior performances in a simulated gridworld environment and a real mice behavior dataset. They analyze HIQL's performances in terms of predicting the expert's behavior and recovering the expert's intention transition dynamics. In the end, they show some evidence of recovering exploration behavior on a mice reversal-learning task.

**Audience:**

Yes

**Claims And Evidence:**

Yes

**Requested Changes:**

Please see the weaknesses for recommended changes.

**Strengths And Weaknesses:**

Strengths:
* The ideas are well-motivated and novel. Modeling animals' intentions across time with a Markov process is intuitive.
* Overall the paper is well written. The story and theoretical justification are clear.
* The experiments clearly show that the proposed method is better than the baseline DIRL.

Weaknesses
* Notations in Sec 4 and Figures should be more consistent:
    * The predicted sequence of reward functions are represented by z1,...,zn in the text but r1,...,rn in Fig 1
    * What is P in Fig 1?
    * Why is delta^(K-1) but R^K
* Some details of the experiments are missing:
    * In Fig 3c, how are the two state value functions(goal and abandon) calculated in the 1 intention/1 map case?
    * In the Gridworld benchmark, how many trajectories are used to evaluate the policies?
    * What's the runtime of this algorithm?
    * It would be clearer if P('tired') in Fig 6d and 5d share the same color.
* It would be helpful if authors discuss why HIQL is better than DIRL even in the 1 intention case.
* Please clarify what's 'win-stay' and 'lose-switch' in sec 5.3.
* It is a little unsatisfying that in three of the experiments, there are only 2 3 intentions for the agents.
* An in-depth analysis of why HIQL is so much better than DIRL would help readers intuitively understand the advantages of this method.
* Are the prediction errors for DIRL mainly happening during intention transitioning? Or is DIRL having a hard time predicting the correct intention overall?

---

> ### Author Response · Authors · 2024-07-31
>
> We thank reviewer NGQK for the questions and suggestions to improve our submission. We address your concerns in detail below.
>
> `“The predicted sequence of reward functions are represented by z1,...,zn in the text but r1,...,rn in Fig 1”`
>
> Here the $r_1, \ldots, r_n$ indicate the reward functions under which the actions $a_1, \ldots, a_n$ were selected, while the $z_1, \ldots, z_n$ denote the corresponding indexes of the reward functions. We suppose that directly depicting the reward function instances ($r_1, \ldots, r_n$) in the figure, instead of their indexes ($z_1, \ldots, z_n$), would be more straightforward for the understanding of the assumed hierarchical decision making process and align better with those commonly used graphical representations of MDPs (e.g., Figure 34.12 from [Murphy, K. P. (2023). Probabilistic machine learning: Advanced topics. MIT press.]).
>
> `“What is P in Fig 1?”`
>
> According to our definition of MDPs in Section 3, The function $P \colon \mathcal{S} \times \mathcal{A} \times \mathcal{S} \to [0, 1]$ in Figure 1 is the state transition function with $P(s, a, s') = \mathbf{P}(s' \mid s, a)$ and $\mathbf{1}^T P(s, a, \cdot) = 1$.
>
> `“Why is delta^(K-1) but R^K”`
>
> Here we take the definition of probability simplex from [Boyd, S., & Vandenberghe, L. (2004). Convex optimization. Cambridge university press.] that $\Delta^{K-1}$ denotes a standard simplex or probability simplex in $\mathbf{R}^K$ whose vertices are the $K$ standard unit vectors in $\mathbf{R}^K$, or in other words $\Delta^{K-1} \triangleq \lbrace x \in \mathbf{R}^K \mid x \succeq 0, \mathbf{1}^T x = 1\rbrace$.
>
> `“In Fig 3c, how are the two state value functions(goal and abandon) calculated in the 1 intention/1 map case?”`
>
> Since EVD is defined as the mean square error between the state-value under the true reward function for the **expert** policy and the **learnt** policy, we thus depicted the state-value under the true reward for the expert policy in the first column of Figure 3c, and the state-value under the true reward for the learnt policy in the other columns of Figure 3c. Under the mult-intention setup, the two learnt policy was evaluated under the corresponding true reward individually to provide the state-value for the two rows in the figure. Under the single-intention setup, since only one learnt policy could be provided by the algorithm, the same output learnt policy was evaluated **twice** repeatedly under the different true reward functions respectively in order to provide the two rows in the figure. We have also included the above information in the current revision.
>
> `“In the Gridworld benchmark, how many trajectories are used to evaluate the policies?”`
>
> The results shown in Figure 3b and 3c were generated based on the learnt policy using the whole dataset with 1024 trajectories (with cross-validation applied). We have included this information in the latest revision.
>
> `“What's the runtime of this algorithm?”`
>
> HIAVI is based on the EM approach, which consists of an inner loop of solving an inverse RL problem (using IQL), and an outer loop that optimizes over the hidden variables (intentions for each step). Kalweit et al. (2020) proposed that the computational complexity of IQL is akin to that of standard value iteration under model-based scenario, with the difference being an additive term for solving the SLE – a term that is often of minor importance given the usual predominance of the number of states over the number of actions. In their work, they also provided some empirical results about the running time of IQL in different environments. However, regarding the outer loop, the number of iterations required for convergence can vary depending on the initial parameter estimates and the convergence criteria (e.g., tolerance levels for change in log-likelihood or parameter values). Hence, the general running time of HIAVI cannot be universally compared without context. In comparison with DIRL, which is also an iterative method with an inner loop for solving the inverse RL problem using MaxEnt IRL, and an outer loop optimizing over the reward mixing weights, HIAVI has a reduction on the inner loop computing complexity (Kalweit et al. 2020). Thus we suppose that under a similar condition for the outer loop convergence criteria, HIAVI would have a shorter running time than DIRL. (Although such criteria would not be trivial to find since the outer loop of HIAVI and DIRL aims to solve the classification problem (about the intention labels) and the regression problem (about the reward mixing weights), respectively, which makes a fair empirical comparison infeasible.)
>
> `“It would be clearer if P('tired') in Fig 6d and 5d share the same color.”`
>
> Thank you for your suggestion, we have modified Figure 6d so that the $\mathbf{P}(\mbox{`tired’})$ has the same color as that in Figure 5d.

---

> > ### Author Response · Authors · 2024-07-31
> >
> > `“It would be helpful if authors discuss why HIQL is better than DIRL even in the 1 intention case.”`
> >
> > Under the single intention setup, HIQL and DIRL fall back to IAVI and MaxEnt IRL, respectively. Hence, the reason why HIQL performs better than DIRL in learning the unknown expert reward under the single-intention scenario is probably that IAVI learns the unknown expert reward in closed-form by solving a system of linear equations about the unknown reward instead of performing parameter estimation as maximum entropy IRL does, which introduces approximate error (Kalweit et al., 2020). We have included this discussion in the current revision.
> >
> > `“Please clarify what's 'win-stay' and 'lose-switch' in sec 5.3.”`
> >
> > We take the definition of ‘win-stay’ and ‘lose-switch’ strategy from those used in [Ashwood et al. Nat. Neuroscience 25.2 (2022): 201-212., Hattori et al. Cell, 2019, 177(7): 1858-1872. e15., Hamaguchi et al. PNAS, 2022, 119(48): e2206067119.], which says ‘stay on the same side for the next trial if it is rewarded in the current trial, and switch to the other one otherwise’. We have now included this information in the current manuscript revision.
> >
> > `“It is a little unsatisfying that in three of the experiments, there are only 2 3 intentions for the agents.”`
> >
> > The proposed HIQL algorithm in this work aims at providing an interpretable representation about animal and human behavior, where the number of intentions that an animal subject may have would be in general limited (Ashwood et al. Nat. Neuroscience (2022)). On the other hand, our method does not have a constraint on the total number of intentions that the expert might have during its demonstration, meaning that in theory HIQL is able to be applied to environments where the expert has unlimited number of intentions (with the cost of running time). Nevertheless, we propose that it would be more valuable and have better interpretability to learn a hierarchical intention model under more complex environments, instead of just simply increasing the number of intentions, which is above the claims in this paper but we believe will provide meaningful directions for future work.
> >
> > `“An in-depth analysis of why HIQL is so much better than DIRL would help readers intuitively understand the advantages of this method.”`
> >
> > Thanks for your suggestion, we have added the discussion about this point in section 6.
> >
> > `“Are the prediction errors for DIRL mainly happening during intention transitioning? Or is DIRL having a hard time predicting the correct intention overall?”`
> >
> > The prediction errors of DIRL under our experiments are mainly because this method cannot properly fit to the discrete intention transition dynamics due to its smoothly varying reward assumption. As a result, in the gridworld benchmark (section 5.1), DIRL had trouble in learning the correct expert reward functions and the policy, i.e., it could not predict the correct intentions. In the real-world mice navigation benchmark (section 5.2), according to Ashwood et al. NeurIPS (2022), the learnt expert policies under two intentions provided by DIRL were similar to those provided by HIQL, indicating that in this task DIRL was able to detect the intentions. However, the recovered intention transition dynamics from DIRL (cf. Figure 4 of Ashwood et al. NeurIPS (2022)) seems not to be as clear and convincing as that provided by DIRL (Figure 5d and 6d), which might lead to the overall poor performance in predicting mice’s behavior.
> >
> > Does this address your concerns?

---

> > > ### Comment · Reviewer_NGQK · 2024-08-17
> > >
> > > Thank you for the detailed response. I believe all my concerns are addressed.

---

### Author Response · Authors · 2024-07-31
**Overview of revision**

We thank all the reviewers for their helpful and insightful comments. In the revised version of the paper, we have addressed the reviewers’ concerns. All updated sections are written in blue in the current revision of the paper.

* Section 3: a more detailed description about IAVI and IQL algorithms from Kalweit et al. (2020) is now included; corresponding algorithms (Algorithm 1 and 2) are listed in Appendix B.
* Section 4: more technical details about HIQL are included; a more formal pseudocode about HIQL (Algorithm 3) is updated and moved to Appendix B.
* Section 5:
    * Some details about the datasets used for the experiments are now added to the main body of the paper, including the number of trajectories used for the evaluation, source of the dataset, etc.
    * Section 5.1: a more detailed discussion about why HIAVI outperformed DIRL even under the single-intention setup is added; the process of calculating the state-values and EVDs (Figure 3c) under single-intention setup is now described in more detail.
    * Section 5.2: the BIC values for DIRL are added to Figure 5b and 6b with corresponding discussion included.
* Section 6: discussion about the main advantages of HIQL compared to the state-of-the-art DIRL is now included.

---

### Comment · Action_Editor_88M8 · 2024-08-02
**Reminder: reviewer - author dicsussion**

Dear reviewers,

the authors have swiftly responded to the reviews and uploaded a revised manuscript. Please read the other reviews and the authors' responses, as well as the changes made in the revised manuscript (Open review lets you easily compare changes across revisions). Raise any outstanding questions or issues before the discussion period ends. In any case, please indicate to the authors that you have read their response.

Thank you for your effort and engagement!
  AE

---

### Decision · Action_Editor_88M8 · 2024-08-27

**Recommendation:** Accept as is

**Comment:**

Taking all information available, I think the paper is novel and original, makes concrete scientific claims that are well supported by evidence, and is ready and interesting to be shared with the TMLR audience. All reviewers agree with this assessment and recommend acceptance as is. No issues were brought up in the reviewer-AE discussion (since there was nothing left to discuss after the rebuttal).

This is a clear case with all information available already in the reviews and authors' responses. I do not have anything to add and clearly recommend accepting the paper.

**Audience:**

The paper is interesting to a large part of the TMLR audience, which reviewers also clearly agree with.

**Claims And Evidence:**

Already in their initial reviews all reviewers agreed that all claims made in the submission are supported by accurate and convincing evidence. Reviewers have confirmed this again in their final recommendations, and I fully agree.